# HONEY: HARMONIZING PROGRESSIVE FEDERATED LEARNING VIA ELASTIC SYNERGY ACROSS DIFFERENT TRAINING BLOCKS

## ABSTRACT

Memory limitation is becoming the prevailing challenge that hinders the deployment of Federated Learning on mobile/IoT devices in real-world cases. Progressive training offers a promising alternative to surpass memory constraints. Instead of updating the full model in each training round, progressive training divides the model into multiple blocks and iteratively updates each block until the full model is converged. However, existing progressive training approaches suffer from prominent accuracy degradation as training each block in isolation drives it to prioritize features that are only beneficial to its specific needs, neglecting the overall learning objective. To address this issue, we present **Honey**, a synergistic progressive training approach that integrates the holistic view and block-wise feedback to facilitate the training of each block. Specifically, the holistic view broadens the learning scope of each block, ensuring that it operates in harmony with the global objective and benefits the training of the whole model. Simultaneously, block-wise feedback heightens each block's awareness of its role and position within the full model, empowering it to make real-time adjustments based on insights from downstream blocks and facilitating a smooth and consistent information flow. Furthermore, to fully harness the heterogeneous memory resources of participating devices, we develop an elastic resource harmonization protocol. This protocol authorizes each device to adaptively train specific layers according to their memory capacity, optimizing resource utilization, sparking cross-block communication, and accelerating model convergence. Comprehensive experiments on benchmark datasets and models demonstrate that **Honey** outperforms state-of-the-art approaches, delivering an exceptional average accuracy improvement of up to 43.9%. Moreover, **Honey** achieves comparable performance even with a reduction in peak memory usage of up to 49%.

## 1 INTRODUCTION

Federated Learning (FL) (McMahan et al., 2017; Wang et al., 2023) is a distributed learning paradigm that enables multiple mobile and IoT devices to collaboratively train a shared model while preserving data privacy. Despite the promising benefits, memory limitation of the participating devices becomes the fundamental and prevailing challenge that hinders the deployment of FL in real-world cases. Due to the intensive memory footprint of the local training process, the low-end devices cannot contribute to the shared model with their own private data (Zhan et al., 2024). Several works have been proposed to surmount the resource limitation, which can be mainly divided into the following two categories: 1) model-heterogeneous training and 2) partial training. Model-heterogeneous training (Li & Wang, 2019; Itahara et al., 2021) customizes local models based on the memory capacity of devices, employing a high-quality public dataset for model aggregation. However, such public datasets are frequently hard to retrieve due to privacy concerns. Partial training tailors the global model through *width scaling* (Diao et al., 2020; Alam et al., 2022) or *depth scaling* (Kim et al., 2022; Liu et al., 2022), and then allocates the sub-models accordingly. However, *width scaling* can compromise the model architecture, and *depth scaling* restricts the complexity of the global model that can be trained.

Recently, progressive training (Wu et al., 2024c) offers a promising alternative to break the memory wall for FL. Unlike traditional FL algorithms (Tian et al., 2024; Ning et al., 2024), progressive training strategically segments the global model into blocks and trains them in a progressive manner. Specifically, the process starts by training the first block. Once it converges, this block is frozen, and the training of the next one is triggered (Wu et al., 2024b). This procedure iterates until the full model is comprehensively trained. In this way, dedicating each round to training a single block effectively reduces the memory footprint while addressing the challenges of model-heterogeneous training and partial training, as it eliminates the need for a shared dataset, preserves the integrity of the model architecture, and places no limitations on the complexity of the global model.

However, progressive training suffers from performance degradation as training each block in isolation restricts its awareness of subsequent blocks, leading to a narrow and short-sighted learning scope (Wang et al., 2021). Due to their limited fitting capacity, these blocks tend to extract features that satisfy their immediate training needs, neglecting the overarching learning objective. This oversight results in a significant loss of valuable information. Consequently, the subsequent blocks experience *accuracy stagnation* during training (see details in Appendix A.2), struggling to learn more insightful features because they have to build on a weakened and information-deficient feature set. Previous efforts in progressive training primarily concentrate on designing local loss functions (Wu et al., 2024b) or developing new training paradigms (Wu et al., 2024c) to assist each block in learning the expected feature representation. Nonetheless, these approaches still fail to recognize the importance of strengthening collaboration between blocks.

Inspired by the above observations, we hypothesize that aligning each block's training objective is promising to be a rescue for progressive training. Therefore, we propose **Honey**, a synergistic progressive training approach that fuses holistic view and block-wise feedback to promote each block's operation in harmony with the global objective while reinforcing block collaboration. Specifically, for each block, we infuse the global training objective and the impact of the current block's updates on downstream blocks into its training objective. These strategies facilitate the model to extract features in a hierarchical and collaborative manner, fostering a smooth and consistent information flow. Furthermore, confining each device to train only its designated block results in a considerable waste of valuable resources. To address this issue, we propose an elastic resource harmonization protocol, which empowers devices to dynamically choose the number of layers to train according to their memory capacity. This protocol optimizes resource utilization and breaks gradient isolation between blocks, cultivating a more resilient training ecosystem.

Comprehensive empirical results demonstrate that **Honey** outperforms existing memory-efficient baselines and progressive training approaches on representative datasets, including CIFAR10 (Krizhevsky et al., 2009), CIFAR100 (Krizhevsky et al., 2009), SVHN (Netzer et al., 2011), STL-10 (Coates et al., 2011), and Tiny-ImageNet (Le & Yang, 2015). Moreover, **Honey** achieves comparable performance even with a reduction in peak memory usage of up to 49%.

## 2 MOTIVATION

### 2.1 THE MEMORY WALL HINDERS THE DEPLOYMENT OF FL

In this section, we aim to explore the question: *how does the memory wall impact the deployment of FL?* Specifically, we establish a pool consisting of 100 mobile devices and distribute the CIFAR10 and CIFAR100 datasets among them in a Non-IID manner. The Non-IID partitioning follows the Dirichlet distribution (Hsu et al., 2019) with a concentration parameter $\alpha = 1$, and ResNet18 is employed as the global model. In each training round, 10% of the de-

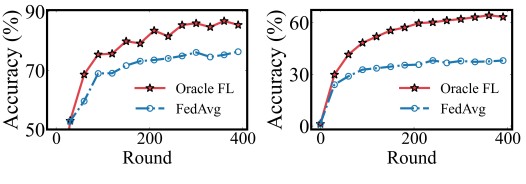

(a) CIFAR10 (Non-IID). (b) CIFAR100 (Non-IID).

Figure 1: Training ResNet18 on CIFAR10 and CIFAR100 datasets in real-world cases.

vices are randomly selected to participate. We adopt the same memory distribution as NeuLite (Wu et al., 2024b) to simulate real-world conditions and employ the FedAvg (McMahan et al., 2017) to execute the FL process. For benchmarking, we also evaluate the performance of *Oracle FL*, which serves as a theoretical baseline and assumes that all the participating devices have sufficient memory resources. Figure 1 presents the experimental results, revealing a noticeable performance decline

in FedAvg compared to *Oracle FL*. For example, on the CIFAR100 dataset, FedAvg experiences a 24.8% accuracy reduction. This is because the memory wall restricts many low-memory devices from participating in FL. These results highlight the critical challenge posed by the memory wall, hindering the successful deployment of FL in real-world scenarios.

## 2.2 EXPLORING EXISTING APPROACHES

In this section, we examine the existing work in resource-aware FL and quantitatively analyze their deficiency. HeteroFL (Diao et al., 2020) and FedRolex (Alam et al., 2022), both landmark approaches designed to address memory constraints in FL, are considered the foremost state-of-the-art methods in the field. These approaches scale the number of channels in convolutional layers according to various criteria, as discussed in Section 5.1, extracting submodels of varying complexity to accommodate the memory constraints of devices. We adopt the same experimental setup as described in Fe-

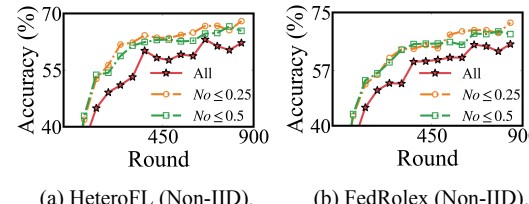

(a) HeteroFL (Non-IID).  (b) FedRolex (Non-IID).

Figure 2: Performance evaluation of different methods in FL on CIFAR10. $No \leq \dagger$ means that devices with a capacity less than or equal to $\dagger$ will not participate in FL, while *All* denotes participation by all devices.

dRolex (Alam et al., 2022). Additionally, to simulate memory heterogeneity, we randomly assign values from the set {1, 0.75, 0.5, 0.25, 0.125} to devices, representing the model complexity each device can handle (Alam et al., 2022). For example, 0.5 indicates that the device can only train half of the channels in each layer of the global model.

To investigate how these approaches compromise the model architecture, we apply different thresholds $\dagger$ to determine which devices participate in training. For instance, $No \leq 0.5$ excludes devices with a memory capacity of 0.5 or below from the training process. Figure 2 shows the experimental results. Interestingly, excluding low-capacity devices does not undermine the model's performance; in fact, it leads to improvements. For example, in Figure 2 (b), the accuracy of $No \leq 0.5$ surpasses that of including all devices (*All*) by 1.5%, even though 60% of the devices are excluded from the FL process. This suggests that these methods struggle to effectively utilize data from low-memory devices, and such a partitioning strategy may even compromise the model architecture, degrading model performance. More experimental analyses are provided in Appendix A.1. Therefore, there is an urgent need to develop more effective methods to break the memory wall in FL.

## 3 PROGRESSIVE TRAINING IN FL

Building on the above motivations, we seek to break the memory wall in FL from a new perspective—progressive training. In this section, we begin with a brief introduction to progressive training, followed by an explanation of why a straightforward implementation falls short.

## 3.1 BACKGROUND

Given a global model $\Theta$, the central server initially divides $\Theta$ into $T$ ($[\theta_1, \theta_2, ..., \theta_T]$) blocks, with each block corresponding to a specific training stage. We define the operational function for block $\theta_t$ as $f^{\theta_t}(\cdot)$, with $\theta_{t,F}$ representing the corresponding frozen block. All blocks, except the last one, are concatenated with an output module $\theta_{op}$ to facilitate independent training. The progressive training process, illustrated in Figure 3 (b), primarily consists of the following key steps (Wu et al., 2024b): 1) **Model Assembly:** The central server assembles the global sub-model $\Theta_t([\theta_{1,F}, \theta_{2,F}, ..., \theta_t, \theta_{op}])$ for the current stage $t$, starting with stage 1. 2) **Device Selection:** The server then selects a subset of devices $S$ to participate in the training round, based on their memory capacity to ensure they can handle the training of the global sub-model $\Theta_t$. The assembled sub-model is subsequently distributed to the selected devices. 3) **Local Training:** The selected devices conduct local training on their private datasets and then upload the updated model parameters ($[\theta_t, \theta_{op}]$) back to the server. 4) **Convergence Assessment:** The server aggregates these updates and evaluates whether the current block has converged. 5) **Model Growing**: Upon achieving convergence, the server freezes the converged block $\theta_t$ and concatenates a new block $\theta_{t+1}$, constructing the sub-model for the next stage. These steps iterate until all blocks are fully trained. Compared to vanilla FL, as shown in

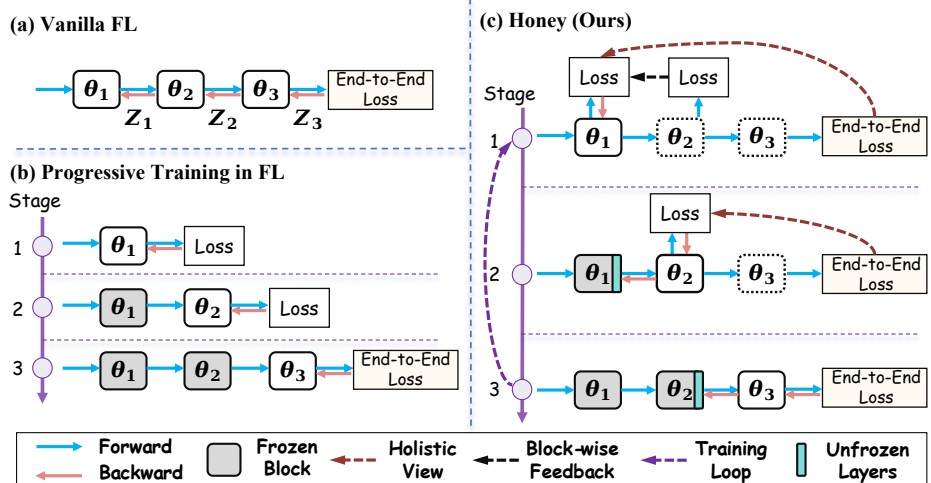

Figure 3: The local training process on the device side. (a) and (b) illustrate the paradigms of vanilla FL and progressive training in FL, where the global model is divided into three blocks. In vanilla FL, all three blocks are updated simultaneously based on the "end-to-end loss". Conversely, in progressive training, all blocks except the last one are trained according to their specific training objectives. Once the block in the current stage converges, freeze it and concatenate a new block, progressing to the next training stage. (c) presents the workflow of **Honey**, where, during the update of each block, in addition to the local training objective, holistic view—specifically, the "end-to-end loss"—and block-wise feedback are incorporated. Additionally, each device unfreezes previously frozen layers according to its memory capacity, known as *unfrozen layers*, and trains them alongside the current block. Moreover, model growth is performed in each training round.

Figure 3 (a), which continuously trains the full model in an end-to-end manner, progressive training concentrates on training one block in each round, notably reducing the memory footprint.

### 3.2 Challenges in Progressive Training

Though achieving memory reduction, existing progressive training approaches suffer from performance degradation, especially when dividing the model into multiple blocks. The following experiments are conducted to investigate this issue. Specifically, we experiment with dividing the ResNet18 into $T$ ($T \in \{1, 2, 4, 8\}$) blocks on CIFAR10, where $T = 1$ represents *Oracle FL*. The experimental results, illustrated in Figure 4, clearly reveal that *model accuracy decreases as the number of blocks increases*. For instance, in the Non-IID scenario, accuracy drops by 4% when the model is divided into two blocks ($T = 2$) compared to $T = 1$. Moreover, this performance decline becomes even more pronounced when the model is divided into eight blocks ($T = 8$), resulting in a significant 21.7% reduction in accuracy. This is because training each block in isolation drives it to extract features that are only beneficial to its specific needs, ignoring the existence and needs of the subsequent blocks. This narrow and short-sighted learning scope can result in the loss of valuable information during the training of earlier blocks, preventing subsequent blocks from extracting more discriminative features. As a result, the later blocks experience *accuracy stagnation*,

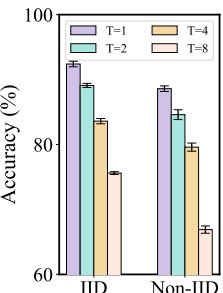

Figure 4: Testing accuracy of ResNet18 on CIFAR10. The global model is segmented into $T$ blocks.

ultimately undermining the model's overall performance. Therefore, naively performing progressive training is insufficient. More analyses from the perspective of information theory are provided in Appendix A.2. To address this challenge, we propose **Honey**, which integrates the holistic view and block-wise feedback to overcome short-sightedness and strengthen block collaboration.

## 4 Honey: A synergistic Progressive Training Approach

Figure 3 (c) presents the workflow of **Honey**. In this section, we first introduce how to infuse the holistic view into each block, steering its learning process. Then, a block-wise feedback mechanism

is developed to guarantee that each block remains aware of the subsequent blocks, empowering it to better serve their needs and fostering a more cohesive training pipeline. Finally, we design an elastic resource harmonization protocol, which not only optimizes resource utilization but also breaks gradient isolation between blocks, boosting training efficiency and model performance.

## 4.1 LEARNING WITH HOLISTIC VIEW

The input to block $\theta_t$ is the output from the preceding block, denoted as $Z_{t-1}$. After passing through $\theta_t$, $Z_{t-1}$ transforms into $Z_t$, which is represented by $Z_t = f^{\theta_t}(Z_{t-1})$. Subsequently, $Z_t$ is processed by the output module $\theta_{op}$, compared with the labels to calculate the empirical loss, $L_t$, which is defined as $L_t = L(f^{\theta_{op}}(Z_t), Y)$. The block $\theta_t$ is then updated according to Eq. (1), where $\theta_t^k$ denotes the parameters from the $k$-th iteration and $\eta$ refers to learning rate.

$$\theta_t^{k+1} \leftarrow \theta_t^k - \eta \cdot \frac{\partial L_t}{\partial \theta_t} \tag{1}$$

Updating block $\theta_t$ solely based on Eq. (1) is like looking through a keyhole—narrow and short-sighted (Wang et al., 2021)—missing the bigger picture that includes subsequent blocks ($[\theta_{t+1}, \theta_{t+2}, ..., \theta_T]$). This myopic learning mechanism drives each block to focus on its immediate objective, extracting features that may seem beneficial for its own performance but potentially undermine the model's overall performance. To counteract this shortsightedness, we infuse the holistic view into the update objective of each block, ensuring its update direction aligns well with the full model. Specifically, we define $Z_T$ as the output of the final block and compute the end-to-end loss with the labels as $L_T$. By integrating $L_T$ into the training objective of block $\theta_t$, we redefine its update target as $L_t + \gamma_t \cdot L_T$, where $\gamma_t$ serves as a weighting factor, striking a balance between the local objective of each block and the global objective of the model. Consequently, the update process for block $\theta_t$ can be expressed as:

$$\theta_t^{k+1} \leftarrow \theta_t^k - \eta \cdot \frac{\partial(L_t + \gamma_t \cdot L_T)}{\partial \theta_t} \tag{2}$$

This updating strategy guarantees that each block works in harmony with the global objective. In this manner, each block not only optimizes its own performance but also contributes positively to the overall model's success, enhancing the coherence and effectiveness of the model's learning process.

## 4.2 LEARNING WITH BLOCK-WISE FEEDBACK

While incorporating the holistic view through Eq. (2) provides a strategy to update block $\theta_t$, a critical challenge remains unaddressed: the transition from the output of block $\theta_t$, $Z_t$, to the input of the final block, $Z_{T-1}$. This transition process remains a black box, offering limited insights and control over the intermediate representation. To bridge this gap, we propose a more refined updating mechanism by incorporating block-wise feedback from downstream blocks into the training objective of each block. This mechanism empowers block $\theta_t$ to be aware of the subsequent blocks and adjust its behavior in response to the needs of downstream blocks, thereby exercising greater control over the transformation of information as it propagates through the network. Specifically, the overall update objective of block $\theta_t$ and the detailed update process are outlined as follows:

$$L_t^{overall} = L_t + \beta_t \cdot (L_{t+1} + L_{t+2} + \cdots + L_{T-1}) + \gamma_t \cdot L_T \tag{3}$$

$$\theta_t^{k+1} \leftarrow \theta_t^k - \eta \cdot \frac{\partial \left( L_t^{overall} \right)}{\partial \theta_t} \tag{4}$$

where $\beta_t$ is a hyperparameter that serves to balance the contribution of block-wise feedback. In this way, block $\theta_t$ not only broadens its learning scope through the holistic view but also gains awareness of its specific role and position within the overall model via block-wise feedback. This awareness enables block $\theta_t$ to make real-time adjustments, responding dynamically to feedback from downstream blocks. As a result, each block is better equipped to adapt its behavior to align with the evolving demands of the full model. This updating mechanism creates a more fluid and responsive training pipeline, promoting deeper synergy and collaboration among blocks. Furthermore, by facilitating communication and alignment across blocks, each block contributes more effectively to the model's unified objective. Thus, block-wise feedback transforms the learning process from a series of isolated updates into a coordinated effort, optimizing the network's performance systematically.

## 4.3 ELASTIC RESOURCE HARMONIZATION

Integrating the holistic view and block-wise feedback greatly improves the training process of each block. However, progressive training still encounters a significant challenge: the failure to fully capitalize on the heterogeneous memory resources of participating devices. This shortcoming arises because, in each training stage, all devices are restricted to training the same block, leading to the underutilization of high-end devices with larger memory capacity. To address this inefficiency, the elastic resource harmonization protocol is developed.

This protocol allows each device to extend its training efforts beyond the current training block $\theta_t$, depending on its available memory and processing capacity. Specifically, devices with more resources can break through and unfreeze previously frozen layers, known as *unfrozen layers* and denoted as $L_{break}$. These layers are updated simultaneously with block $\theta_t$ during training. By dynamically unfreezing and training these layers, we secure that each device makes the most of its capabilities, thus optimizing resource utilization and enhancing the overall training efficiency. In this way, we cultivate a more efficient and scalable FL training ecosystem where each device actively advances the model's global objective. The complete update process can be formulated as follows:

$$\theta_t^{k+1} + \theta_{L_{break}}^{k+1} \leftarrow (\theta_t^k + \theta_{L_{break}}^k) - \eta \cdot \frac{\partial(L_t^{overall})}{\partial(\theta_t + \theta_{L_{break}})} \tag{5}$$

After completing local training, devices only need to upload their updated model parameters to the central server for aggregation, effectively reducing communication overhead. To prevent high-end devices from dominating the aggregation process, we design a balanced aggregation strategy. Particularly, layers that are unfrozen and updated on high-end devices are aggregated with the corresponding frozen layers from low-end devices using the FedAvg (McMahan et al., 2017). This aggregation strategy ensures a fair contribution from all devices, regardless of their resources, promoting a more inclusive and balanced training process. Additionally, as outlined in Eq. (5), both the current block and previous layers are trained together, effectively breaking gradient isolation and enhancing collaborative adaptation across blocks (Wu et al., 2024b).

## 5 EXPERIMENTS

### 5.1 EXPERIMENTAL SETUP

**Default Settings.** We evaluate the effectiveness of `Honey` using the following representative datasets: CIFAR10 (Krizhevsky et al., 2009), CIFAR100 (Krizhevsky et al., 2009), SVHN (Netzer et al., 2011), STL-10 (Coates et al., 2011), and Tiny-ImageNet (Le & Yang, 2015). Additionally, we employ models from three popular architectures—namely, ResNet (He et al., 2016), VGG (Simonyan & Zisserman, 2014), and Transformer (Vaswani et al., 2017)—as global models. The datasets are partitioned in both IID and Non-IID forms among 100 devices, except for STL-10, distributed among 20 devices. The Non-IID distribution is based on the Dirichlet distribution (Hsu et al., 2019) with $\alpha = 1$. In each training round, 10% of the devices are randomly selected to participate, except for STL-10, where 20% are selected. We use the same memory settings as NeuLite (Wu et al., 2024b), based on profiling results from various mobile devices. The details are presented in Appendix A.3. During local training, each device performs five local epochs using SGD as the optimizer with a learning rate of 0.01, except for Tiny-ImageNet, which uses AdamW (Loshchilov, 2017) with a learning rate of 0.0001.

**Baselines.** We employ the following baselines for comparison: 1) *AllSmall (Wu et al., 2024c):* A naive baseline that scales down the number of convolutional channels in the global model based on the device with the smallest memory capacity, creating a model that allows all devices to participate in training. 2) *ExclusiveFL (Liu et al., 2022):* This approach restricts participation to devices with enough memory capacity to train the full model, excluding those with insufficient memory from the training process. 3) *DepthFL (Kim et al., 2022):* This method applies *depth scaling* to the global model, creating models with varying depths that are assigned to devices based on their memory capacity. 4) *HeteroFL (Diao et al., 2020):* A static *width scaling* algorithm that adjusts the number of convolutional channels in the global model to obtain sub-models of varying complexity. 5) *FedRolex (Alam et al., 2022):* Similar to HeteroFL but employs a sliding window to extract sub-models. 6) *TiFL (Chai et al., 2020):* This approach stratifies devices based on their training

Table 1: Performance comparison of various FL methods to train different models across different datasets. **Bold** and Underlined indicate the optimal and sub-optimal results, respectively. The − symbol signifies that the corresponding algorithm fails to work under this setup. For progressive training methods, the global model is divided into four blocks based on the model architecture.

| | | CIFAR10 | | | | CIFAR100 | | | | |
| | Method | IID | | Non-IID | | IID | | Non-IID | | Average |
| | | Res18 | Res34 | Res18 | Res34 | Res18 | Res34 | Res18 | Res34 | |
|---|---|---|---|---|---|---|---|---|---|---|
| Basic Approach | AllSmall | 76.8% | 67.0% | 69.5% | 53.8% | 37.5% | 27.4% | 17.4% | 9.4% | 44.9% |
| | ExclusiveFL | 77.9% | - | 76.8% | - | 37.1% | - | 35.2% | - | 28.4% |
| Partial Training | DepthFL | 79.3% | 80.1% | 65.1% | 73.2% | 36.5% | 47.0% | 33.3% | 43.1% | 57.2% |
| | HeteroFL | 82.8% | 9.9% | 76.7% | 10.0% | 47.0% | 1.1% | 34.8% | 1.0% | 32.9% |
| | FedRolex | 84.7% | 81.4% | 76.6% | 71.8% | 51.3% | 44.3% | 35.7% | 26.5% | 59.0% |
| Client Selection | TiFL | 81.0% | - | 73.6% | - | 40.7% | - | 37.6% | - | 29.1% |
| | Oort | 76.9% | - | 75.9% | - | 41.4% | - | 35.3% | - | 28.7% |
| Progressive Training | InfoPro$^S$ | 82.1% | 83.3% | 74.5% | 74.0% | 52.5% | 53.0% | 46.7% | 47.3% | 64.2% |
| | InfoPro$^D$ | 83.5% | 85.0% | 73.3% | 71.9% | 53.4% | 55.0% | 47.3% | 48.9% | 64.8% |
| | SmartFreeze | 82.8% | 82.0% | 76.7% | 72.0% | 54.4% | 50.7% | 48.2% | 46.2% | 64.1% |
| | NeuLite | 87.0% | 84.2% | 80.4% | 74.9% | 57.3% | 54.8% | 51.2% | 49.5% | 67.4% |
| | **Honey** | **89.5%** | **87.5%** | **85.1%** | **82.0%** | **63.3%** | **59.1%** | **57.7%** | **53.8%** | **72.3%** |

time and selects devices from specific tiers for each training round accordingly. 7) *Oort (Lai et al., 2021):* This approach simultaneously accounts for both system and data heterogeneity to select devices. 8) *InfoPro$^S$ (Wang et al., 2021):* Although InfoPro is not specifically designed for FL, we adapt it for FL by introducing an additional reconstruction loss during the training of each block to reduce information loss. Additionally, model growth is triggered when a block has converged. 9) *InfoPro$^D$ (Wang et al., 2021):* This approach adopts a dynamic model growth strategy, where the model grows in each training round. 10) *SmartFreeze (Wu et al., 2024a):* A progressive training algorithm tailors a dedicated output module for each block. 11) *NeuLite (Wu et al., 2024b):* An advanced progressive training approach that customizes the training loss for each block based on information theory and enhances information interaction from both directions.

## 5.2 END-TO-END EVALUATION

In this section, we evaluate the effectiveness of **Honey** from two perspectives: 1) overall performance compared to various baselines, and 2) performance comparison with progressive training approaches under different partitioning schemes.

**Overall Performance.** Table 1 presents the model performance of various methods under different experimental settings. We observe that **Honey** demonstrates significant superiority, with an average accuracy improvement of up to 43.9%. Specifically, when training ResNet18 on CIFAR10 (IID), **Honey** improves accuracy by 12.7% compared to AllSmall. This is because the global model complexity of AllSmall is constrained by the device with the smallest memory capacity, leading to insufficient feature extraction capabilities. **Honey** outperforms ExclusiveFL with an 11.6% increase in accuracy, attributed to its inclusive framework. DepthFL shows a 10.2% decrease in accuracy compared to **Honey** due to its imbalanced parameter training and inability to effectively utilize data from low-memory devices. Compared to *width scaling* methods like HeteroFL and FedRolex, **Honey** achieves performance gains of 6.7% and 4.8%, respectively. This is because *width scaling* compromises the model architecture. Methods like TiFL and Oort experience up to a 12.6% decrease in accuracy because they fail to utilize data from low-memory devices. Compared to InfoPro$^S$ and InfoPro$^D$, **Honey** still achieves performance gains of 7.4% and 6.0%, respectively. These improvements are due to InfoPro's increased memory usage from its complex reconstruction module, which limits its ability to effectively utilize data from low-memory devices. Additionally, it lacks **Honey**'s ability for inter-block collaboration to efficiently extract features. Even compared to other progressive training methods like SmartFreeze and NeuLite, **Honey** improves accuracy by 6.7% and 2.5%, respectively. This is because **Honey** exploits the holistic view and block-wise feedback to guide the training process of each block. At the same time, **Honey** also fully utilizes the heterogeneous memory resources of all devices through elastic resource harmonization.

Table 2: Comparison with progressive training methods. In this set of experiments, memory limitations are not considered, and `Honey` disables the elastic resource harmonization. The ∗ symbol indicates *Oracle FL*, where all devices train the full model end-to-end, serving as the upper bound. ResNet18 is employed as the global model.

| Dataset | Distribution | Method | T=1 | T=2 | T=4 | T=8 |
|---|---|---|---|---|---|---|
| CIFAR10 | IID | SmartFreeze | | 89.1% (↓ 2.0%) | 83.6% (↓ 5.2%) | 75.6% (↓ 10.2%) |
| | | NeuLite | 92.4%∗ | 90.2% (↓ 0.9%) | 85.5% (↓ 3.3%) | 81.8% (↓ 4.0%) |
| | | **Honey** | | **91.1%** | **88.8%** | **85.8%** |
| | Non-IID | SmartFreeze | | 83.6% (↓ 4.7%) | 76.4% (↓ 4.7%) | 65.9% (↓ 12.3%) |
| | | NeuLite | 88.8%∗ | 85.8% (↓ 2.5%) | 79.1% (↓ 2.0%) | 73.9% (↓ 4.3%) |
| | | **Honey** | | **88.3%** | **81.1%** | **78.2%** |
| CIFAR100 | IID | SmartFreeze | | 61.7% (↓ 5.6%) | 56.4% (↓ 5.8%) | 49.8% (↓ 10.3%) |
| | | NeuLite | 68.6%∗ | 63.6% (↓ 3.7%) | 59.7% (↓ 2.5%) | 55.7% (↓ 4.4%) |
| | | **Honey** | | **67.3%** | **62.2%** | **60.1%** |
| | Non-IID | SmartFreeze | | 56.5% (↓ 5.3%) | 50.5% (↓ 5.5%) | 43.7% (↓ 9.4%) |
| | | NeuLite | 61.2%∗ | 59.8% (↓ 2.0%) | 54.6% (↓ 1.4%) | 48.9% (↓ 4.2%) |
| | | **Honey** | | **61.8%** | **56.0%** | **53.1%** |
| SVHN | IID | SmartFreeze | | 93.8% (↑ 0.2%) | 91.1% (↓ 0.9%) | 85.8% (↓ 5.8%) |
| | | NeuLite | 91.9%∗ | **94.2%** (↑ 0.6%) | 91.2% (↓ 0.8%) | 90.1% (↓ 1.5%) |
| | | **Honey** | | 93.6% | **92.0%** | **91.6%** |
| | Non-IID | SmartFreeze | | 92.9% (↑ 0.4%) | 90.2% (↓ 0.8%) | 83.7% (↓ 6.7%) |
| | | NeuLite | 91.7%∗ | **93.3%** (↑ 0.8%) | 89.9% (↓ 1.1%) | 88.4% (↓ 2.0%) |
| | | **Honey** | | 92.5% | **91.0%** | **90.4%** |
| STL-10 | IID | SmartFreeze | | 73.4% (↓ 3.6%) | 68.5% (↓ 4.7%) | 65.8% (↓ 5.1%) |
| | | NeuLite | 77.2%∗ | **77.6%** (↑ 0.6%) | 72.3% (↓ 0.9%) | 69.8% (↓ 1.1%) |
| | | **Honey** | | 77.0% | **73.2%** | **70.9%** |
| | Non-IID | SmartFreeze | | 72.3% (↓ 2.5%) | 65.6% (↓ 3.9%) | 62.3% (↓ 5.3%) |
| | | NeuLite | 75.1%∗ | **76.3%** (↑ 1.5%) | 67.9% (↓ 1.6%) | 64.6% (↓ 3.0%) |
| | | **Honey** | | 74.8% | **69.5%** | **67.6%** |

**Comparison with Progressive Training Methods.** To demonstrate the superiority of `Honey` over other progressive training approaches, we employ different partitioning schemes on the global model and compare their performance. In this set of experiments, we operate under the assumption of no memory constraints, and `Honey`'s elastic resource harmonization protocol is disabled, allowing us to focus on the effectiveness of the holistic view and block-wise feedback during training. This practice is intended to highlight the importance of fostering block collaboration within the model. Specifically, we train ResNet18 on the CIFAR10, CIFAR100, SVHN, and STL-10 datasets, partitioning the global model into $T$ ($T \in \{1, 2, 4, 8\}$) blocks. SmartFreeze and NeuLite serve as baselines, with the experimental results presented in Table 2.

We observe that across all experimental settings, `Honey` consistently achieves near-optimal performance, particularly when the model is divided into more blocks. For example, on the CIFAR10 (IID) dataset, with $T = 2$, `Honey` improves accuracy by 2.0% over SmartFreeze and 0.9% over NeuLite. At $T = 8$, the improvements are even more pronounced, with a 10.2% increase over SmartFreeze and a 4.0% increase over NeuLite. Moreover, on the SVHN dataset, even with a peak memory reduction

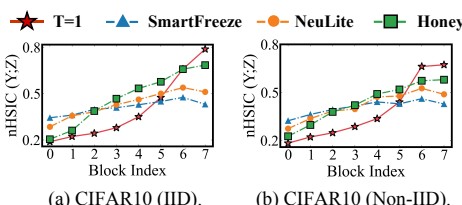

(a) CIFAR10 (IID).  (b) CIFAR10 (Non-IID).

Figure 5: nHSIC(Y; Z) of each block on the CIFAR10 when dividing into eight blocks.

of up to 49% when $T = 8$, `Honey` maintains performance on par with *Oracle FL*. Additionally, for the CIFAR10 dataset at $T = 8$, we freeze the global model parameters trained by each algorithm and compute the nHSIC(Y; Z) (Ma et al., 2020) between each block's output $Z$ and the labels $Y$, as shown in Figure 5. The nHSIC(Y; Z) captures the correlation between the extracted features and the labels. The results show that while SmartFreeze and NeuLite effectively extract essential features in the earlier blocks, information loss prevents the later blocks from capturing more critical features. In contrast, `Honey` successfully mitigates the short-sightedness of these methods, extracting features in a hierarchical fashion similar to end-to-end training ($T = 1$).

## 5.3 MEMORY EFFICIENCY

Despite introducing the holistic view and block-wise feedback, **Honey** remains a memory-efficient approach. Taking ResNet models as an example, we divide them into four and eight blocks and compare the peak memory usage across different methods. In this set of experiments, we also disable the elastic resource harmonization protocol. Figure 6 presents the results on the CIFAR10 with a batch size of 256, where **Honey** shows a significant reduction in peak memory usage compared to *Oracle FL*. For instance,

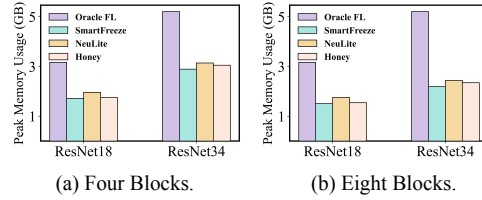

(a) Four Blocks.     (b) Eight Blocks.

Figure 6: Peak memory usage of different methods on the CIFAR10 dataset.

as shown in Figure 6 (b), dividing ResNet34 into eight blocks reduces the peak memory footprint by up to 53%. Compared to other methods, **Honey** incurs a negligible additional memory overhead.

## 5.4 MODEL UNIVERSALITY

In this section, we demonstrate the model universality of **Honey** by training VGG16 on CIFAR100 and Vision Transformer (ViT) (Dosovitskiy et al., 2020) on Tiny-ImageNet. Figure 7 presents the experimental results, showing that **Honey** achieves even superior performance compared to *Oracle FL* ($T = 1$) across various training tasks and partitioning schemes. For example, on the Tiny-ImageNet, **Honey** improves accuracy by 0.9% compared to *Oracle FL* when $T = 4$. This is because optimizing models without skip connections in an end-to-end

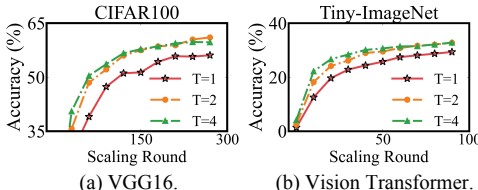

(a) VGG16.     (b) Vision Transformer.

Figure 7: Model Universality. Both datasets are partitioned in a Non-IID manner, with the global models divided into $\{1, 2, 4\}$ blocks.

manner is challenging, whereas **Honey** efficiently trains each block in a progressive manner. More experimental results are provided in Appendix A.4.

## 5.5 SENSITIVITY ANALYSIS

We then perform a sensitivity analysis on the hyperparameters $\beta_t$ and $\gamma_t$, using the ranges $\beta_t \in [0.1, 0.3]$ and $\gamma_t \in [0.2, 0.8]$ as an example. Three strategies are utilized to determine these hyperparameters: 1) Constant-**Honey**$^C$: fixed values of $\beta_t = 0.2$ and $\gamma_t = 0.5$; 2) Gradu-

Table 3: Sensitivity Analysis.

| Dataset | Distribution | **Honey**$^C$ | **Honey**$^I$ | **Honey**$^D$ |
|---|---|---|---|---|
| CIFAR10 | IID | 89.6% | **90.1%** | 89.3% |
| | Non-IID | 84.9% | **85.4%** | **85.4%** |
| CIFAR100 | IID | 63.4% | **64.1%** | 63.7% |
| | Non-IID | 57.2% | **57.5%** | 56.6% |

ally Increasing-**Honey**$^I$: $\beta_t$ and $\gamma_t$ linearly increase with block index; 3) Gradually Decreasing-**Honey**$^D$: $\beta_t$ and $\gamma_t$ linearly decrease with block index. ResNet18 is employed as the global model. The results, summarized in Table 3, demonstrate that **Honey** exhibits robustness to hyperparameter selection. Notably, the strategy of gradually increasing $\beta_t$ and $\gamma_t$ yields the best performance.

## 5.6 ABLATION STUDY

We further conduct a breakdown analysis of the benefit brought by each component, i.e., the holistic view, block-wise feedback, and elastic resource harmonization. Experimental results, as shown in Table 4, indicate that each component makes a significant contribution to performance improvement. For example, eliminating the holistic view leads to an average accuracy decrease of 1.3%, while removing block-wise feedback causes an average drop of 0.7%. Furthermore, omitting elastic resource harmonization results in an average accuracy decline of 1.9%.

## 6 RELATED WORK

**Model-heterogeneous training** involves customizing local models of different complexity for participating devices according to their memory capacity (Itahara et al., 2021; Zhang et al., 2022; Lin

Table 4: Ablation Study. $w/o\ HV$ denotes without the holistic view, $w/o\ BF$ represents the omission of block-wise feedback, and $w/o\ ERH$ indicates removing the elastic resource harmonization.

| Method | CIFAR10 | | | | CIFAR100 | | | | Average |
|--------|---------|---|------|---|---------|---|------|---|---------|
| | IID | | Non-IID | | IID | | Non-IID | | |
| | Res18 | Res34 | Res18 | Res34 | Res18 | Res34 | Res18 | Res34 | |
| *w/o HV* | 87.9% | 86.7% | 83.5% | 81.4% | 62.3% | 58.1% | 56.1% | 51.6% | 71.0% (↓ 1.3%) |
| *w/o BF* | 89.3% | 87.2% | 84.6% | 80.9% | 63.1% | 57.8% | 56.9% | 52.9% | 71.6% (↓ 0.7%) |
| *w/o ERH* | 88.2% | 87.2% | 82.6% | 78.8% | 61.1% | 58.3% | 54.7% | 52.1% | 70.4% (↓ 1.9%) |
| **Honey** | **89.5%** | **87.5%** | **85.1%** | **82.0%** | **63.3%** | **59.1%** | **57.7%** | **53.8%** | **72.3%** |

et al., 2020), with knowledge distillation (Hinton et al., 2015) used for aggregation across different model architectures. For instance, in FedMD (Li & Wang, 2019), devices upload the logits computed on a shared dataset to facilitate knowledge transfer after completing local training. Similarly, Fed-ET (Cho et al., 2022) employs a data-aware weighted consensus distillation on a public dataset to transfer the knowledge from an ensemble of models to the server model. However, retrieving such public datasets is typically challenging due to data privacy concerns.

**Partial training** encompasses techniques that employ *width scaling* or *depth scaling* on the global model, producing sub-models of varying complexity. HeteroFL (Diao et al., 2020), a well-established *width scaling* approach, scales the number of convolutional channels in the global model based on the memory capacity of devices, statically extracting sub-models of different complexity. Unlike HeteroFL, FedRolex (Alam et al., 2022) employs a sliding window to dynamically extract sub-models. However, this strategy compromises the model architecture, leading to performance degradation. Conversely, InclusiveFL (Liu et al., 2022) and DepthFL (Kim et al., 2022) are representative *depth scaling* methods that address memory limitations by constructing models of varying depths. However, these methods typically assume that some devices have sufficient memory to train the full model, an assumption that is challenging to meet in real-world scenarios.

**Progressive training** is a new learning paradigm that divides the global model into blocks and trains them in a progressive manner to reduce memory usage during training. However, this paradigm typically drives each block to learn features that only benefit itself, overlooking the overall model performance. To address this challenge, SmartFreeze (Wu et al., 2024a) constructs corresponding output modules for each block, enabling it to be aware of subsequent blocks. ProFL (Wu et al., 2024c) decouples model training into two stages, assisting each block in learning the expected feature representation. Meanwhile, NeuLite (Wu et al., 2024b), from the perspective of information bottleneck theory (Ma et al., 2020), designs specialized training losses for each block and breaks information isolation across blocks in both forward and backward directions. However, these methods still fail to recognize the importance of fostering collaboration between blocks and struggle to fully harness the heterogeneous memory resources of participating devices.

# 7 CONCLUSION

In this paper, we propose **Honey**, a synergistic progressive training approach to break the memory wall for FL. To assist the training process of each block, we integrate the holistic view and block-wise feedback into its training objective. Specifically, the holistic view ensures that each block operates in harmony with the global objective and contributes positively to the overall model's success. Simultaneously, block-wise feedback strengthens block collaboration, facilitating a smooth and consistent information flow. Furthermore, to optimize the utilization of heterogeneous memory resources of participating devices, we propose an elastic resource harmonization protocol. This protocol enables devices to adaptively train specific layers according to their memory capacity, thereby accelerating model convergence and sparking cross-block communication. Our comprehensive experiments on representative datasets and models demonstrate that **Honey** outperforms existing memory-efficient methods by a large margin and achieves accuracy comparable to *Oracle FL* even with a reduction in peak memory usage of up to 49%. It is worth noting that **Honey** requires a full forward propagation during the training of each block, which introduces some additional computation overhead. Minimizing this overhead will be a primary focus of our future work.

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

# A APPENDIX

## A.1 LIMITATIONS OF WIDTH SCALING APPROACHES

Devices participating in FL typically operate with limited resources (Li et al., 2023; Tian et al., 2022; Li et al., 2022; Wu et al., 2023), making it crucial to account for resource constraints when designing algorithms. To thoroughly assess the feasibility of existing *width scaling* methodologies, these approaches are directly applied to the local training process on each device. The experimental setup is as follows: The CIFAR10 dataset is distributed across 100 devices, with 10% of devices randomly selected to participate in each training round, and ResNet18 is employed as the global model. HeteroFL (Diao et al., 2020), Federated Dropout (Caldas et al., 2018), and FedRolex (Alam et al., 2022) are selected for evaluation. These algorithms scale the number of channels in the convolutional layers based on different criteria to generate sub-models of varying complexity, thereby meeting diverse memory constraints. Specifically, HeteroFL employs a static approach, Federated Dropout utilizes a random approach, and FedRolex adopts a sliding window to extract sub-models. Figure 8 illustrates two rounds of these approaches on two participating clients with heterogeneous memory capacity. To mimic the scenario of memory heterogeneity, we randomly allocate a number from the set $\{1, 0.75, 0.5, 0.25, 0.125\}$ to devices, representing the model complexity that can be trained. Furthermore, we assess the effectiveness of these methods across three key aspects: the impact of device capacity, the influence of global model complexity, and the effect of high-capacity devices.

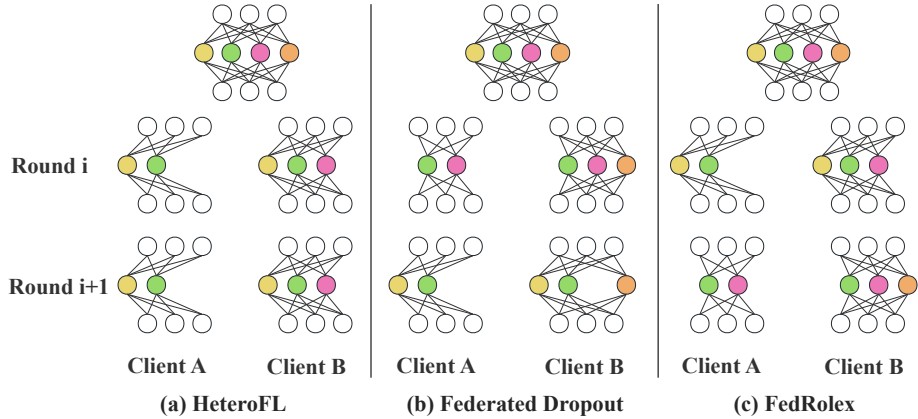

Figure 8: Illustration of how sub-models are extracted by different sub-model extraction schemes. (a) HeteroFL: static sub-model extraction scheme. (b) Federated Dropout: random sub-model extraction scheme. (c) FedRolex: rolling sub-model extraction scheme.

Table 5 presents the experimental results of the three algorithms under various settings. Regarding the effect of device capacity on global model performance, it is clear that excluding low-capacity devices does not negatively impact the global model's performance and may even lead to improvements. For example, in the case of HeteroFL under Non-IID conditions, $No \leq 0.25$ improves accuracy by 2.22% compared to including all devices (*All*). This suggests that these algorithms struggle to effectively leverage data from low-capacity devices, and such partitioning strategies may even disrupt the model architecture, thereby compromising model performance.

To explore the effect of global model complexity, we conduct experiments by reducing the global model's complexity to allow all devices to participate in FL. In this set of experiments, we assume that all devices' capacity is aligned with the model's complexity. For instance, G(0.5) indicates that the global model's complexity is reduced to half of its original size. It can be observed that reducing the complexity of the global model significantly compromises the model's performance. For example, when the global model's complexity is halved, HeteroFL experiences a 7.26% drop in accuracy under Non-IID conditions. This degradation stems from the diminished feature extraction capability due to the reduced model complexity, as well as the disruption of the model architecture caused by *width scaling*. Furthermore, to assess the effectiveness of these algorithms in more realistic FL scenarios where no devices possess sufficient memory to train the full model, we evaluate

Table 5: Performance evaluation of different memory optimization algorithms in FL on CIFAR10. $No \leq \dagger$ means that devices with a capacity less than or equal to $\dagger$ will not participate in FL. $G(\dagger)$ means that the complexity of the global model is $\dagger$ times that of ResNet18. $FD$ stands for the Federated Dropout and **Bold** indicates the optimal results.

| Distribution | Method | Device Capacity Effect | | | | | Global Model Effect | | | High-Capacity Effect |
| --- | --- | --- | --- | --- | --- | --- | --- | --- | --- | --- |
| | | All | $No \leq 0.125$ | $No \leq 0.25$ | $No \leq 0.5$ | $No \leq 0.75$ | G (1) | G (0.75) | G (0.5) | No capacity 1 |
| IID | HeteroFL | **82.23%** | 81.75% | 82.20% | 81.47% | 79.90% | **88.76%** | 87.37% | 84.17% | 30.99%(-51.24%) |
| | FD | 80.24% | 81.28% | 82.07% | **83.25%** | 79.01% | **87.01%** | 84.64% | 80.85% | 74.91%(-5.33%) |
| | FedRolex | 83.39% | 84.23% | **84.73%** | 84.50% | 79.63% | **88.81%** | 87.14% | 84.39% | 80.16%(-3.23%) |
| Non-IID | HeteroFL | 63.18% | 64.26% | **65.40%** | 64.26% | 61.75% | **76.87%** | 74.81% | 69.61% | 8.60%(-54.58%) |
| | FD | 42.82% | 55.05% | 59.59% | **63.47%** | 59.79% | **66.95%** | 65.77% | 62.67% | 31.78%(-11.04%) |
| | FedRolex | 66.23% | 67.13% | **70.39%** | 67.68% | 61.04% | **76.98%** | 74.51% | 70.19% | 47.89%(-18.34%) |

cases where no device has a capacity of 1. As shown in Table 5, regarding the high-capacity effect, we observe that these algorithms perform poorly under such scenarios. Notably, HeteroFL suffers a substantial 54.58% accuracy drop under Non-IID conditions. This sharp decline occurs because, without devices capable of training the full model, certain channels are not adequately trained. In conclusion, *width scaling* algorithms fail to effectively address the memory constraints in FL.

## A.2 ANALYSIS OF THE PROGRESSIVE TRAINING PARADIGM

To explore the underlying causes of performance degradation resulting from progressive training, we employ a strategy similar to those used in InfoPro (Wang et al., 2021) and NeuLite (Wu et al., 2024b) to analyze the feature representation learned by each block under different block division schemes. Specifically, we concentrate on three key metrics: testing accuracy, nHSIC(X; Z) (Ma et al., 2020), and nHSIC(Y; Z) (Ma et al., 2020), to gain deeper insights into the learning dynamics. Testing accuracy indicates the linear separability of the learned features (Wang et al., 2021), nHSIC(X; Z) measures the amount of information about the input $X$ contained in the activations $Z$, and nHSIC(Y; Z) reflects the correlation between the learned features and the labels. Additionally, to evaluate the testing accuracy of each block, we freeze the global model parameters obtained from training under different block division schemes. We then attach an output module to each block to perform linear probing (Kumar et al., 2022).

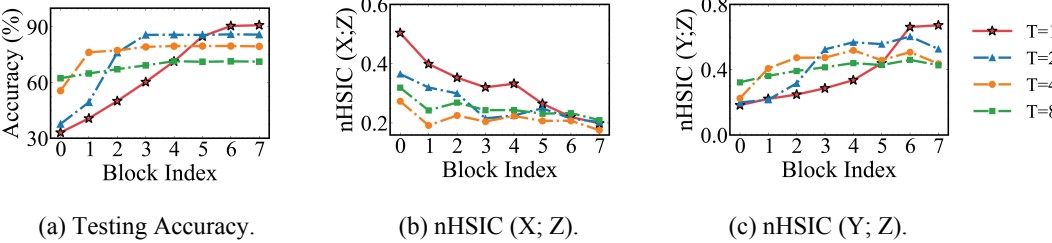

(a) Testing Accuracy.     (b) nHSIC (X; Z).     (c) nHSIC (Y; Z).

Figure 9: Training ResNet18 on the CIFAR10 dataset under different block division schemes. (a): Testing accuracy of each block. (b): nHSIC(X; Z) of each block indicates the amount of input information contained in the feature representation extracted by each block. (c): nHSIC(Y; Z) of each block measures how effectively the features capture label-relevant information.

Figure 9 (a) illustrates the testing accuracy of each block, with the X-axis denoting the block index and the Y-axis representing the testing accuracy achieved via linear probing for the corresponding block. Interestingly, when $T = 1$, we observe a progressive improvement in accuracy across the blocks, suggesting effective collaboration among them to capture critical features. However, for $T > 1$, although the initial blocks achieve higher accuracy, the overall model performance is lower compared to $T = 1$. For example, with $T = 2$, even though the accuracy at block index 4 improves by 25.4% compared to $T = 1$, the overall model accuracy still declines by 5.1%. This downward trend becomes more pronounced as $T$ increases. When $T$ reaches 8, the performance drop reaches up to 19.6%. Figure 9 (b) shows the nHSIC(X; Z) values for each block. For $T = 1$, nHSIC(X; Z) decreases relatively slowly as the block index increases, indicating that input information is preserved more effectively. In contrast, for $T > 1$, a significant amount of input information is lost

in the earlier blocks, which likely leads to the *accuracy stagnation* observed in the later blocks in Figure 9 (a). This phenomenon indicates that dividing the model into more blocks worsens the loss of valuable input information, ultimately compromising overall performance. Figure 9 (c) illustrates the nHSIC(Y; Z) values for each block, revealing a pattern similar to the trend in testing accuracy. When $T = 1$, each block progressively extracts more discriminative features relevant to the target $Y$. However, when $T > 1$, the blocks are short-sighted, greedily learning features most beneficial to their immediate training objectives while neglecting the overall model goal. This narrow perspective ultimately prevents them from extracting more critical features. Therefore, enhancing collaboration among blocks is essential for effective feature extraction.

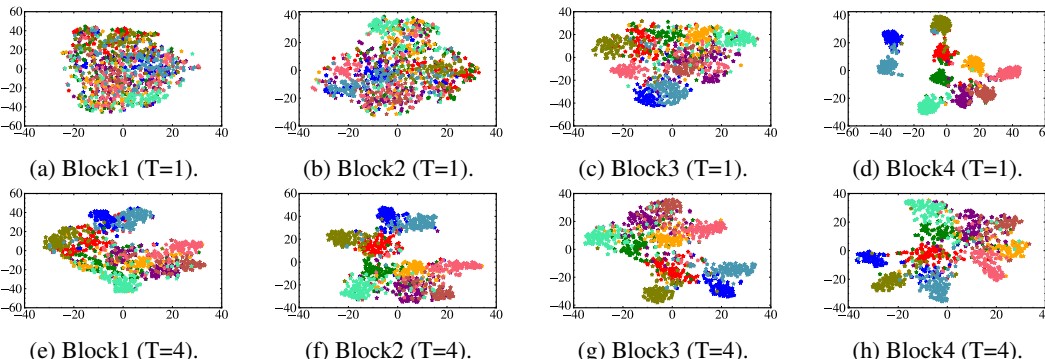

| (a) Block1 (T=1). | (b) Block2 (T=1). | (c) Block3 (T=1). | (d) Block4 (T=1). |
|---|---|---|---|
| (e) Block1 (T=4). | (f) Block2 (T=4). | (g) Block3 (T=4). | (h) Block4 (T=4). |

Figure 10: Visualization results of intermediate activations for each block under both *Oracle FL* ($T = 1$) and progressive training ($T = 4$) in FL on CIFAR10 (Non-IID). Different colored asterisks represent different classes.

We further employ T-SNE (Van der Maaten & Hinton, 2008) to visualize the intermediate activations of each block under both *Oracle FL* ($T = 1$) and progressive training ($T = 4$), as depicted in Figure 10. We observe that in *Oracle FL*, input data is progressively segmented, with each block learning specific feature representation. Conversely, progressive training ($T = 4$) separates input data in a more isolated, greedy manner. Although subsequent blocks in progressive training refine the features extracted by earlier ones, they fail to achieve the performance of *Oracle FL*.

These observations highlight two key insights: 1) The model extracts features in a hierarchical fashion, building complexity layer by layer. 2) Dividing the model into multiple blocks drives each block to concentrate on learning features that are most beneficial for its training objective, ignoring the existence and needs of the subsequent blocks. This narrow learning scope can lose valuable information, undermining the model's performance. Therefore, for each block, embracing a broader view and strengthening collaboration among blocks is essential.

## A.3 EXPERIMENTAL SETUP

We adopt the same memory settings as NeuLite (Wu et al., 2024b). The device participation rates of different methods across various training tasks are shown in Tables 6 and 7.

Table 6: The device participation rate of different methods across various training tasks.

| Method | AllSmall | ExclusiveFL | DepthFL | HeteroFL | FedRolex |
|---|---|---|---|---|---|
| ResNet18 | 100% | 18% | 43% | 100% | 100% |
| ResNet34 | 100% | 0% | 36% | 100% | 100% |

Table 7: The device participation rate of different methods across various training tasks.

| Method | TiFL | Oort | InfoPro$^S$ | InfoPro$^D$ | SmartFreeze | NeuLite | Honey |
|---|---|---|---|---|---|---|---|
| ResNet18 | 18% | 18% | 100% | 100% | 100% | 100% | 100% |
| ResNet34 | 0% | 0% | 100% | 100% | 100% | 100% | 100% |

## A.4 MODEL UNIVERSALITY

In this section, we present the experimental results under IID conditions, as shown in Figure 11. We observe that **Honey** achieves superior performance compared to *Oracle FL*, regardless of whether the global model is divided into two or four blocks. For example, in Figure 11 (a), dividing VGG16 into two blocks results in a 2.9% performance improvement over *Oracle FL*, while dividing it into four blocks leads to a 2.8% improvement. This is because optimizing models without skip connections in an end-to-end manner is challenging, often leading to issues such as vanishing gradients. In contrast, **Honey** effectively trains each block in a progressive manner. Not only does **Honey** reduce memory footprint during training, but it also achieves comparable performance, which is a remarkable achievement.

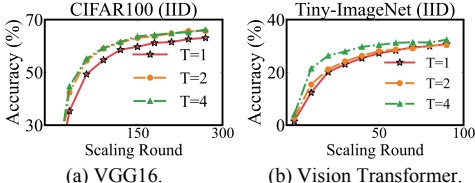

(a) VGG16.  (b) Vision Transformer.

Figure 11: Model Universality. Both datasets are partitioned in a IID manner, with the global models divided into $\{1, 2, 4\}$ blocks.

