# OpenReview forum: "Honey: Harmonizing Progressive Federated Learning via Elastic Synergy across Different Training Blocks"
_ICLR.cc/2025/Conference — ICLR 2025 Conference Withdrawn Submission_

### Official Review · Reviewer_WGhc · 2024-10-26

**Soundness:** 2
**Presentation:** 3
**Contribution:** 2
**Rating:** 3
**Confidence:** 4

**Summary:**

This paper presents Honey, a memory-efficient Federated Learning (FL) paradigm that overcomes the memory constraints of edge devices. Firstly, Honey deploys a progressive training scheme that gradually trains each block in a bottom-first order. Second, Honey utilizes holistic view and block-wise feedback to enhance the collaboration between different blocks. Thirdly, Honey introduces an elastic training strategy that allows clients with higher resources to train more layers to improve performance.

**Strengths:**

1. This paper is well-written and presented.
2. This work of this paper is meaningful. In practice, addressing the memory shortage issue of edge devices is critical in the area of FL.
3. Experiment results show that Honey outperforms SOTA methods in most scenarios.

**Weaknesses:**

1. The ideas of Honey lack novelty and are not quite significant. First, the idea of bottom-first progressive training in Honey has already been explored in [1]. The only difference between Honey and [1] is that, in each stage, Honey trains one block fully and freezes the rest, and [1] trains one block fully and partially trains other blocks. Second, the ideas of holistic view and block-wise feedback are highly similar to that in DepthFL [2]. Specifically, both Honey and DepthFL add a loss function to the end of each block in training, and distill knowledge between blocks by comparing their outputs. The only distinct point is that Honey combines the between-block losses with different coefficients. Based on these two points, it seems that the work of Honey is just a combination of different components from [1] and [2] with slight modification, which lacks original contents and is non-innovative. Lastly, the idea of elastic resource harmonization is too simple, as it just lets clients with more resources unfreeze more layers in training, which is insufficient to be listed as a contribution.

2. It is unclear how the output module $\theta_{op}$ (Section 3.1) is implemented. The authors are supposed to briefly introduce the architecture of the module. For example, is it a fully-connected layer or an external block? Furthermore, adding additional output modules to the original model might increase model complexity, causing larger memory consumption in training. In correspondence,  the authors should analyze the potential negative impact of the output modules on memory efficiency and provide solutions if possible.

3. The authors didn’t show any comparison between the computation and communication costs in the experiment. Since Honey trains a model progressively, it might require more rounds (thus more costs) to reach a target accuracy than traditional methods that consistently train the whole model. As a remedy, the authors are expected to exhibit comparisons between the computation and communication costs in the experiment, such as FLOPs, training time, size of parameter transmission and energy consumption in the experiment.

**Questions:**

1. The experiment baselines are selected inappropriately. It seems that this paper does not involve any new client selection strategy in FL. In this case, why did the authors add client selection methods (TiFL and Oort) to the baselines? Additionally, as a typical progressive training method, [1] is not included for comparison. I recommend the authors add this method to the experiment.


2. The elastic resource harmonization in Honey may incur asynchronous training progress among clients with various computation resources. In communication, the heterogeneous clients may contribute blocks that are trained unevenly to the server.  As a result, the convergence speed of the aggregated global model can be hindered. In this case, can the authors provide some convergence analysis (e.g. how the global gradient changes, how the global objective function decays) to demonstrate the effect of elastic resource harmonization on the global model’s convergence in detail?


3. It is unclear how the memory usage in Section 5.3 is acquired. The authors should explain how they measure or calculate the memory footprint.


[1] Kilian Pfeiffer, Ramin Khalili, and Jörg Henkel. Aggregating capacity in FL through successive layer training for computationally-constrained devices. In Proceedings of the 37th International Conference on Neural Information Processing Systems, 2023.

[2] Kim, Minjae, et al. "DepthFL: Depthwise federated learning for heterogeneous clients." The Eleventh International Conference on Learning Representations. 2023.

---

### Official Review · Reviewer_1qy2 · 2024-10-27

**Soundness:** 2
**Presentation:** 4
**Contribution:** 2
**Rating:** 5
**Confidence:** 3

**Summary:**

This paper proposes a progressive training method called Honey for memory-constrained federated learning. Specifically, to address the block isolation issue, Honey incorporates global objectives and downstream blocks' feedback into the local block training through combined loss; to address the device memory heterogeneity issue, Honey allows each device to unfreeze previously frozen layers according to their available recourses. Honey is evaluated on four datasets in both IID and Non-IID settings, and results show that compared with other progressive training methods, Honey can improve model performance and reduce peak memory usage.

**Strengths:**

1. This paper is well-written, well-organized, and easy to follow.
2. The motivations of each method are well explained.
3. The experimental results of Honey are good.

**Weaknesses:**

1. There are many similarities between this paper and another paper titled "NeuLite: Memory-Efficient Federated Learning via Elastic Progressive Training" in terms of methods and organizations, as well as the one titled "Heterogeneity-Aware Memory Efficient Federated Learning via Progressive Layer Freezing." The authors do not clearly clarify the main improvements, differences, and relationships between this paper and the two publications. Thus, readers may not have a good understanding of their contributions.
2. Each method seems a little bit simple and straightforward, thus the novelty is limited.
3. There is no theoretical analysis of Honey to explain why the proposed methods can outperform other optimized progressive training methods regarding model performance and memory usage.
4. This paper lacks a detailed discussion about the limitations of Honey, e.g., the extra computation complexity.
5. The evaluation part lacks experimental details, e.g., how each device's memory capacities are configured and the default configurations for the two hyper-parameters gamma_t and beta_t. The authors claim they use the same memory settings as NeuLite, but I didn't find detailed settings in NeuLite. And more memory results should be provided.

**Questions:**

1. Since there are many similarities between Honey and NeuLite/SmartFreeze, what are the main differences,  improvements, and relationships between Honey and NeuLite/SmartFreeze?
2. How many computation complexities will Honey introduce? Are such computation consumptions worthwhile?
3. Since Honey introduces two hyper-parameters /gamma_t and /beta_t, how to select the proper values? What (default) configurations are used in each experiment? Will different values affect the performance a lot? Why? Section 5.5 is not enough.
4. Since the authors focus on the memory usage issue, how are the memory capacities of each device configured? Will such capacities change over time? How many clients benefit from the elastic resource harmonization protocol in each round?
5. How many rounds/time are needed for each method and each block to converge? How to determine the block achieves convergence?
6. Since Honey optimizes the loss function, can the authors provide some loss curves over the learning process? Can we get some observations from curves or other results to illustrate that Honey empowers each block "to be aware of the subsequent blocks and adjust its behavior in response to the needs of downstream blocks"?
7. Are each experiment executed several times? Can the authors provide the deviation values to show the stability?
8. The theoretical analysis should be added to explain why Honey outperforms other progressive training methods.
9. More limitations and future work of Honey should be discussed.

---

### Official Review · Reviewer_qhTp · 2024-10-31

**Soundness:** 2
**Presentation:** 2
**Contribution:** 1
**Rating:** 3
**Confidence:** 5

**Summary:**

The paper studies federated learning over heterogenous, memory-limited, edge devices. This is an important problem which attracted lots of attentions recently. The paper however fails to recognize state of the art in this area, and lacks novelty.

**Strengths:**

The problem is important and timely. The ablation study of appendix A2 provides interesting insight. Also, the proposed solution can be applied to a wide range of models, including vision transformers.

**Weaknesses:**

Progressive training is not a new concept and has been explored in several prior works (before Wu et al. 2024a, 2024b, and 2024c), such as Hettinger et al. [1], which progressively adds layers on top of each other and utilizes auxiliary heads throughout training while freezing early layers. Similar progressive training approaches with early exits have also been investigated for unsupervised learning [2,3].

Beyond centralized training, progressive training has been applied in federated learning as well, such as in ProgFed [4] and Fednet2net [5]. Notably, progressive training has been employed to address heterogeneous memory constraints of devices in FL [6]. In [6], freezing early layers is combined with a downscaled head that is progressively increased in width, achieving up to an 8x memory reduction compared to end-to-end training. Since layers are not stacked with auxiliary heads but instead reuse the same head, blocks are not trained in isolation. As this is a primary contribution of the proposed Honey technique, the authors should clarify why this effect is not achieved with prior methods like [6].

The paper also misses out on related work in the domain of partial training within federated learning, particularly studies that apply freezing techniques in federated settings, such as [7,8,9], which share similarities with the proposed approach. For instance, in [8], which also tackles heterogeneous memory constraints affecting computation and communication, only the middle blocks are trained while the early layers and head remain frozen. In this setup, the cross-entropy loss is backpropagated through the frozen head to the trained layers, similar to the approach proposed here (introduced as $L_T$). Thus, the primary novelty of the proposed method in this submission lies in combining the loss from the last layer with that from the early exit.

So, please discuss on how Honey differs from or improves upon these existing approaches, particularly [6] and [8], clarify the novelty, and provide a comparison of Honey's performance against these related works.

Even setting aside these related works, the contribution remains minimal compared to those referenced in the paper (e.g., SmartFreeze and NeuLite). The current submission offers only modifications to these existing techniques, which does not sufficiently justify a new publication.

Section 4 also includes claims that are not supported by any of the conducted experiments. For example, the authors assert that their aggregation scheme “promotes a more inclusive and balanced training process.” However, there are no experiments demonstrating that devices with more limited memory can significantly impact the global model, nor is fairness evaluated. So please provide experimental evidence to support the claim about the inclusivity and balance of the aggregation scheme, e.g. by conducting experiments that demonstrate the impact of devices with limited memory on the global model, and include an evaluation of fairness.

Additionally, Honey requires a full forward pass through the whole model, yet the paper lacks discussion on how this can be achieved without increasing the memory and computation footprint. If there is an increase in memory or computation, quantify and discuss this trade-off, and compare the memory and computation requirements of Honey with other progressive training approaches.

Furthermore, details on memory configuration and device profiling are missing, with Appendix A3 referring readers to the NeuLite paper. The paper should be self-containing, so please provide any key details from the NeuLite paper that are necessary for understanding this approach.



[1] Chris Hettinger, Tanner Christensen, Ben Ehlert, Jeffrey Humpherys, Tyler Jarvis, and Sean Wad: Forward thinking: Building and training neural networks one layer at a time. NeurIPS 2017.

[2] Sindy Löwe, Peter O’Connor, and Bastiaan Veeling. Putting an end to end-to-end: Gradient-isolated learning of representations. NeurIPS 2019.

[3] Yuwen Xiong, Mengye Ren, and Raquel Urtasun. Loco: Local contrastive representation learning. NeurIPS 2020.

[4] Hui-Po Wang, Sebastian Stich, Yang He, and Mario Fritz. Progfed: effective, communication, and computation efficient federated learning by progressive training. In International Conference on Machine Learning. PMLR 2022.

[5] Amit Kumar Kundu and Joseph Jaja. Fednet2net: Saving communication and computations in federated learning with model growing. ICANN 2022.

[6] Kilian Pfeiffer, Ramin Khalili, Jörg Henkel: Aggregating Capacity in FL through Successive Layer Training for Computationally-Constrained Devices. NeurIPS 23.

[7] Tien-Ju Yang, Dhruv Guliani, Françoise Beaufays, and Giovanni Motta. Partial variable training for efficient on-device federated learning. ICASSP 2022.

[8] Kilian Pfeiffer, Martin Rapp, Ramin Khalili, Jörg Henkel: CoCoFL: Communication- and Computation-Aware Federated Learning via Partial NN Freezing and Quantization. TMLR 2023.

[9] Mehdi Setayesh, Xiaoxiao Li, Vincent W.S. WongPerFedMask: Personalized Federated Learning with Optimized Masking Vectors. ICLR 2023.

**Questions:**

How does Honey perform compared with SOTA discussed in the weaknesses section?

---

### Official Review · Reviewer_b3f6 · 2024-11-04

**Soundness:** 3
**Presentation:** 3
**Contribution:** 2
**Rating:** 5
**Confidence:** 4

**Summary:**

The paper addresses the critical challenge of memory limitations in Federated Learning (FL) on mobile/IoT devices. The authors propose a novel approach called Honey, which employs progressive training to divide the model into blocks and update them iteratively.

Honey incorporates a holistic view and block-wise feedback to ensure that each block aligns with the global learning objective and facilitates information flow across blocks. An elastic resource harmonization protocol is also introduced to optimize resource utilization across devices with heterogeneous memory capacities.

The paper claims significant improvements in accuracy and memory efficiency over state-of-the-art methods.

**Strengths:**

The paper introduces a progressive federated learning method that combines holistic and block-wise feedback, enhancing inter-block synergy and memory efficiency across devices with varying capacities. Through an elastic resource harmonization protocol, it optimizes device memory use while significantly improving model accuracy. Comprehensive experiments across different datasets and architectures, along with ablation studies, confirm Honey’s robustness and efficacy.

**Weaknesses:**

This paper has some weaknesses in terms of experimental setup and formatting：

1. Evaluating Honey solely on image classification tasks limits its potential as a general paradigm for federated learning. To better demonstrate Honey’s versatility, it should be tested on a broader range of tasks, such as text classification tasks (like GLUE) or text generation tasks (like SQuAD), and with models of more diverse scales, from traditional small models (like LSTM, CNN) to popular large models (like GPT, BERT).

2. The abstract should be more concise to allow readers to quickly grasp the paper’s content.

3. The captions under some figures are overly lengthy. Detailed explanations should be in the main text rather than in the figure captions.

**Questions:**

1. The choice of using the simplest aggregation method, FedAvg, over more complex alternatives requires further justification. To strengthen this section, it may be necessary to provide more experimental evidence supporting this choice. We recommend that the authors compare FedAvg with one or two specific alternative aggregation methods, such as FedProx or SCAFFOLD. This comparison would provide valuable insights into how Honey’s performance might be influenced by different aggregation strategies and help clarify whether more advanced aggregation methods could further enhance its performance.

2. Honey requires a full forward pass during the training of each block, which may introduce additional computational overhead. What's the effect of the additional computational overhead? Is it tolerable on devices with limited resources? We recommend that the authors do quantative measurements of the additional computational cost on different device types and how this compares to the memory savings achieved by Honey.

---

### Note · Authors · 2024-11-13

I have read and agree with the venue's withdrawal policy on behalf of myself and my co-authors.